# Occupational and Environmental Risk Factors Influencing the Inducement of Erythema among Nigerian Laboratory University Workers with Multiple Chemical Exposures

**DOI:** 10.3390/ijerph16081334

**Published:** 2019-04-13

**Authors:** Usaku Reuben, Ahmad F. Ismail, Abdul L. Ahmad, Humphrey M. Maina, Aziah Daud

**Affiliations:** 1Department of Community Medicine, School of Medical Sciences, Universiti Sains Malaysia; Kubang Kerian 16150, Malaysia; usakureuben5@gmail.com (U.R.); afilza@usm.my (A.F.I.); 2School of Chemical Engineering, Universiti Sains Malaysia, Nibong Tebal Penang 14300, Malaysia; chlatif@usm.my; 3Department of Chemistry, School of Pure and Applied Sciences, Modibbo Adama University of Technology, Yola P.M.B 2076, Adamawa State, Nigeria; humphaks@gmail.com

**Keywords:** erythema, laboratory workers, chemicals

## Abstract

The chemicals from laboratories pose a significant risk forinducing erythema, an abnormal redness of the skin, as a result of poor occupational and environmental factors that promote hypersensitivity to a chemical agent. The aim of this present study was to determine the occupational and environmental risk factors influencing the inducement of erythema in laboratory workers due to exposure to chemicals. This was a cross-sectional study on a population-based sample of Nigerian university laboratory workers. Data were collected using the erythema index meter and an indoor air control meter. The study included 287 laboratory workers. The laboratory workers who properly used personal protective equipment (PPE) were 60% less likely to have induced erythema (adjusted odds ratio (AOR) = 0.40; 95% confidence interval CI: 0.22–0.77; probability value *p* = 0.011). The chemical mixture exceeding the permissible exposure limit (PEL) was found to have a small effect in inducing the erythema (AOR = 4.22; 95%CI: 2.88–12.11; *p* = 0.004). Most of the sampled laboratories where the respondents worked had unsuitable temperatures (AOR = 8.21; 95% CI: 4.03–15.01; *p* = 0.001). Erythema was more frequently found in the respondents who spent 4–5h in the laboratory (AOR = 3.11; 95%CI: 1.77–9.23; *p* = 0.001). However, high levels of ventilation reduce the likelihood of erythema in a laboratory by 82% (0.18). Multiple logistic regressions revealed that PPE, PEL, exposure time, temperature, and ventilation were the probable predictive factors associated with the inducement of erythema. Providing better educational knowledge and improving the attitude towards hazards and safety in a laboratory would lead to reduced rates of new cases.

## 1. Introduction

In this study, erythema is defined as an abnormal redness of the skin caused by hypersensitivity to the chemical agents that are absorbed from the surface to the underlying layers of the human skin. These chemical agents are classified with respect to occupational and environmental conditions which have been proven to significantly contribute to increased levels of pollutants in indoor air, posing numerous inflammatory skin health challenges to laboratory workers and the environment. Indoor air refers to the air quality in and around laboratory buildings and facilities, which influences the health and comfort of the workers when good working process conditions are not put in place [1]. A lack of an integrated legal system in accordance with the Occupational Safety Health and Administration (OSHA) and the International Organization for Standardization (ISO) in Nigerian university laboratories is of much interest and is the major concern of this study. For over two decades, many chemical-related occupational workers in Nigerian universities, especially chemical laboratories, have been injured and left with severe skin health conditions [2]. Furthermore, Jeong et al. [3] indicated that the current attitudes towards hazards and safety as well as poor laboratory practices have contributed to chemical-induced skin injuries and erythema inducement. Chemical-induced injury is commonly experienced when adsorbed haptens penetrate under the skin layers through either dermal absorption or direct contact.

## 2. Cell Cytotoxicity Leading to Erythema Inducement

Toxicity from the effect of chemicals is best described by a mechanism called cell-mediated cytotoxicity. In this study, it is defined as a type of cell-mediated cytotoxic reaction that occurs in human skin cells in an exposed environment, involving the display of foreign antigens on the cell surface protein [4,5]. When the skin is being affected by chemicals in an exposed environment, the cell surface protein will recognize the foreign molecules (haptens) under the skin layers when there is a minimal amount of erythema. The participating cells in the reaction will eventually cause the separation of the periciliary skin layers due to the accumulated amount of chemicals present under the skin layers [6]. However, it is also believed that these toxic chemicals cannot initiate and propagate a reaction on their own under the skin layers due to the small size of the chemical agent [7]. This depends on the cell surface proteins in terms of the facilitation of a reaction in order to display and recognize the antigen with an appropriate T-cell [4]. During exposure, the basal cells become swollen due to the presence of the absorbable chemicals under the skin layers. These chemicals subsequently bind to the body proteins and start to look like an antigen or lymphocyte. This process presents the foreign antigen onto the major histocompatibility complex (MHC) class one molecule for recognition. On the other hand, cytotoxic T-lymphocytes(CTLs) are activated when they bind to the antigen on top of MHC1 by the T-cell receptor in a similar manner to the cluster of differentiation(CD8+ and CD3+) cell surface proteins [8]. This leads to the detachment of the periciliary skin layer, which causes blisters and vesicles that induce erythema and other skin inflammatory conditions (tissue and cell damage) [9].

Survey studies, observations, and complaints have revealed that workers spend a significant amount of their working time in chemical laboratories. They hence become affected by a chemical, which causes inflammation in the skin due to the immunological responses to the foreign and self-antigens from the skin surface. For this study, we determined that erythema was induced in chemical laboratory workers that reported chemical-induced injury and had an erythema index difference (EID) ≥0.1, while there was no erythema in workers with EID <0.1. Hence, the objective of this study was to determine the occupational and environmental risk factors that influence the inducement of erythema among Nigerian university laboratory workers due to multiple chemical exposures.

## 3. Materials and Methods

### 3.1. Data Collection

This was a cross-sectional study conducted from November 2016 to May 2017 with 287 Nigerian university laboratory workers. The study area and the respondents were randomly selected from 30 available accredited Nigerian university laboratories. The erythema index meter and indoor air quality control meter were used to measure the erythema index corresponding to the induced erythema and indoor air concentration (environmental conditions) in a dose-dependent manner. Workers who were aged 18 years old and above and had been working at the same place for at least 2yearswere included in this study. We excluded any workers, who had other chemical-related occupations, had a genetic mutation (albinos), or who were receiving medication that could affect normal hemoglobin and melanin levels. Worker volunteers who did not work at chemical laboratories and had no history of skin allergies and induced erythema provided consent to participate in this study and were used as controls for the purpose of calibrating research tools.

Mean, standard deviation (SD), and frequency (%) were provided as descriptive analyses in this study. Also, an independent sample *t*-test was used to determine mean differences for the continuous variables, whereas the chi-squared test of independence (*x*^2^test) was applied for descriptive statistics of frequency and percentage. A logistic regression (to estimate odds ratio at 95% confidence interval CI) was used for examining the associations between the occupational and environmental characteristics in the total study sample. The alpha level of significance was set at *p*<0.05 throughout the study. We ran all data analyses using the International Business Machines-Statistical Package for the Social Sciences (SPSS version 22.0, IBM, Chicago, IL, USA).

The ethics approval of the current study was granted by the Human Research Ethics Committee of the Universiti Sains Malaysia USM (Ref.USM/JEPeM/16090130), the National Health Research Ethics Committee (NHREC) of Nigeria (Ref.NHREC/01/01/2007-28/12/2016), and the West African Bioethics and Collaborative Institutional Training Initiative (CITI) (Ref.ID:5949144). All the protocols of the study were carried out in agreement with good research practice principles as enshrined in the Helsinki Declaration [10].

### 3.2. Measurement Tool

The erythema-induced allergic response was measured using an erythema index meter (EIM), also known as MX18 [11,12], as shown in Figure 1a. The meter was calibrated and equipped with a probe to measure the chemical-induced skin injury corresponding to the erythema index (EI) of the participants in a susceptive environment. The EI in this study is defined as the threshold of epidermal damage that characterized the quantitative measurement of the biophysical characteristics of laboratory workers. The highly sensitive measurement gives values on a broad scale (0–999) on the measurement tools, with 92% sensitivity for erythema classification so that even the smallest changes in color as a result of haptens under the skin layers become traceable. The probe is diminutive and lightweight for easy handling and measurement on all body sites. A probe head ensures constant pressure on the skin enabling exact, reproducible measurements in a susceptible environment.

The probe was pressed against the skin surface of the participating subjects to block the outside light in order to ensure proper assessment of the interaction. The measurements were obtained from the workers prior to the work sessions in the laboratories (pre-exposure) and after the work sessions (post-exposure). On the other hand, the environmental conditions and indoor air quality were measured using an indoor air quality control meter (IAQCM), which is also known as the EXTECH MODEL SD800, and agas dosimeter tube, which is known as NEXTTEQ 7446-09-5. This is shown in Figure 1b.

The dosimeter tube (also known as the NEXTTEQ 7446-09-5 [13] and IAQCM), which contained substances that have a reaction with the gas of interest to produce a color change, was calibrated, configured, and equipped with a sensor to measure different levels of carbon dioxide (CO_2_), carbon monoxide (CO), temperature (°C), and relative humidity (%). However, sulfur dioxide (SO_2_), hydrogen sulfide (H_2_S), and nitrogen dioxide (NO_2_) were measured as time-weighted averages (TWAs) using dosimeter tubes containing substances that have a reaction with the gas of interest. These tubes are equipped with a length-of-stain indication that is proportional to the amount of gas contaminant present in the laboratory, which end with a discrete line of differentiation. The value on the scale that corresponds to the end of the stain length was the concentration of the target gas. The average concentrations of the gases were obtained by dividing the reading by the total length of time that the tube was exposed (expressed in hours) in the laboratory as seen in the expression below:(1)$$\mathrm{Average}\;\mathrm{Concentration}=\left(\frac{\mathrm{Dosimeter}\;\mathrm{tube}\;\mathrm{reading}\;\left(\mathrm{ppm}.\mathrm{hour}\right)}{\mathrm{Sampling}\;\mathrm{time}\;\left(\mathrm{hours}\right)}\right)$$

## 4. Results

The sociodemographic characteristics of the respondents as well as environmental and chemical parameters are summarized in Table 1a–c. The study included 287 respondents, consisting of 122 females and 165 males. The average age of the analyzed participants was 40.2 (range of 18–58) years. The mean age was 38 (5.1) years for females and 43 (8.1) years for males. The mean working experience and monthly income of the laboratory workers were 13.6 (6.2) years and US$246.60 (US$120.80), respectively. The mean indoor air concentrations of the selected 30 chemical laboratories are shown in Figure 1b in a dose-dependent manner. These were determined to be higher than the international permissible exposure limits and Nigerian air quality guidelines, except for CO and CO_2_, which were below the recommended limits. Table 1b shows the environmental parameters of the same chemical laboratories, which include temperature, laboratory ventilation system (LVS), relative humidity, and size of the laboratory. The results revealed that the parameters were below the recommended international standard (ISO) and Consortium of Local Education Authorities for the Provision of Science Services (CLEAPSS) standards, although the LVS was set as moderate at the period of sampling.

The χ^2^ test revealed that personal protective equipment (PPE), permissible exposure limit (PEL), air laboratory temperature (ALT), indoor air quality (IAQ), and laboratory ventilation system (LVS) were significantly associated with erythema inducement (*p* < 0.001, *p* = 0.031, *p* < 0.001, *p* = 0.001, and *p* = 0.002, respectively) (Table 2). Furthermore, an independent sample *t*-test revealed that the exposed population and time of exposure (TOE) were significantly different between these two groups (95% CI; *p* = 0.001 and *p* < 0.001, respectively) (Table 3).

Table 4 shows the final model that was established after the necessary statistical tests. All the variables in the model were statistically significant with a *p*-value less than 0.05. The variables were ordered preferential to the best selection procedure and the preliminary main effect model processed using the enter method. The model accounted for the matching by factors best on the statistically significant variables. The significant variables that were retained in the multivariable logistic regression for determining the associated factors influencing the erythema inducement included PPE (*p* = 0.011), PEL (*p* = 0.004), TOE (*p* = 0.001), ALT (*p* = 0.001), and LVS (*p* = 0.002). In addition, the model stability and the diagnostic ability of the binary classifier shown in Figure 2 were found to be 85.6%, which demonstrates a good fit of the model.

## 5. Discussion

In recent times, many novel bioengineering techniques have been proposed for assessment of patch test for erythematous reactions with a more objective approach. Many studies are aimed at investigating the effectiveness and usefulness of the erythema index in chemical-related occupations and in clinical settings, i.e., interpretation of allergic patch tests and positive associations [14]. The purpose of this study was to determine the occupational and environmental risk factors that influence the inducement of erythema among Nigerian university laboratory workers due to multiple chemical exposures. In addition, there is no doubt that every inflammatory condition leads to erythema inducement and immunological responses as a result of foreign and self-antigens on exposure. Consequently, such a vascular–circulatory condition is undoubtedly present in inflammations of any organs or tissues that have blood capillaries. The current study in its own way provided baseline immunological responses (the erythema index) to foreign and self-antigens among chemical laboratory workers in a susceptive environment.

In this paper, we report that certain occupational and environmental characteristics (specifically PPE, PEL, TOE, temperature, relative humidity, and LVS) were potential risk factors that were associated with the inducement of erythema. The proper use of the polymers of the ethylene vinyl alcohol is an effective safety measure during a work session in a laboratory. Jeong and Kim [3] reported that an estimated 30–45% of all cases of occupational diseases in laboratories are due to an inappropriate use of PPE, with only 10–15% of all laboratory workers properly using PPE during work in the laboratories. According to the National Institute for Occupational Safety and Health (NIOSH) and OSHA, the lack of the proper use of PPE might be associated with a rise in the rate of skin allergic conditions and erythema inducement in laboratories [14]. In this study, the multivariate analysis showed that the proper use of PPE at the time of sampling resulted in the odds of erythema inducement being reduced by 60%.

The permissible exposure limit was reported by Kheur et al. [15] as an independent factor causing skin allergies, with skin biopsies revealing that this occurred when PEL was used in a range of 0.03–13.48 ppm for the soluble group (median of 0.115 ppm). Furthermore, they discovered that the silver concentrations found during work sessions exceeded 0.01 mg/m^3^, which is the threshold limit value (TLV) set by the NIOSH for laboratories. However, this study reported that the chemical mixture with a concentration above the TLV (PLE exceeded) at the time of sampling resulted in significantly higher odds of erythema inducement compared to those who were not exposed to a concentration above the TLV. Thus, the higher odds found in the findings could be due to the high probability that the chemical substance will cause harm under certain conditions of use when the chemical levels exceed the PEL [16]. Another reason accounting for the higher odds and disparity might be the susceptibility of the biological system of the participants in a susceptive environment (the ability of a chemical substance to elicit a toxic response) [17].

In a study conducted in Germany by Geieret al. [18], they revealed that this was associated with the duration of the chemical allergen on the skin. The risk and effect of exposure was proportional to the duration of the exposure in the susceptive environment [19]. Similarly, according to our findings, the workers who spent 4–5 h working with chemical substances in a laboratory were 8.2 times more likely to experience erythema inducement compared to those who worked for 2–3 h after controlling for other variables. This finding could account for the fact that the toxic effects of chemical exposure depend on the amount, type, and length of time of exposure to the harmful substances; most importantly, severity is related to a long duration of exposure [17]. Many factors play a part in whether a subject will be affected by erythema inducement after being in contact with chemicals. The place of origin of the subjects, especially those from countries with a high prevalence of skin allergies, is believed to contribute extensively and could reflect the disparities in the results [20].

Temperature has been reported to be a specific factor for skin allergies that influences erythema inducement in a chemical laboratory [21]. The American Society of Heating, Refrigerating and Air-Conditioner Engineers (ASHRAE) standard was set to regulate the temperature under working conditions in conjunction with OSHA and NIOSH guidelines for both employers and employees. A high working environmental temperature has been documented in Taiwan by Albert and Chang [22] as a good associated factor with the dermal absorption and direct contact with the chemical allergens in laboratories. According to our findings, a temperature greater than 31 °C, which is above the international limit, at the time of sampling significantly increased the likelihood of erythema inducement compared to low temperatures(26.6–31.9 °C). In a chemical laboratory, a high temperature could increase chemo thermal and dermal absorption of haptens under the skin layers as the faster movement of the particles results in them colliding with each other more frequently under high temperatures, hence speeding up the inducement of erythema and other allergic reactions under the skin layers [23,24]. An increased temperature increases the reaction rates of allergic reactions given that the high-energy collisions might increase the cytotoxicity of the cells to an appropriate T-cell [8].

There is growing evidence supporting the inducement of erythema due to the effect of LVS on laboratory workers [25,26]. An expert documented that infiltration, which is the introduction of outside air into a laboratory building that is typically described in terms of the air changes per hour (ACH), is believed to have a high potential in reducing the dermal absorption of the chemicals under the skin layers [27]. For the descriptive purpose of the study, this factor was subcategorized into poor (≤20 cfm^−1^), moderate (21.5–40.5 cfm^−1^), and good (≥41.5 cfm^−1^) LVS. According to our findings, a highly ventilated laboratory at the time of sampling (expressed as LVS ≥ 41.5 cfm^−1^, believed to be above the international limit) was 82% (0.18 times) less likely to cause erythema inducement and other skin allergic conditions compared to an environment that had an LVS of more than 20 cfm^−1^. These findings suggest that an efficient LVS supplies and removes air through diffusers or vents that are strategically located in the laboratories. A high local ventilation system, such as a chemical fume hood, will remove more toxic substances or pollutants from the point of generation in the laboratory. The face velocity measurements might be a result of this trend. In the OSHA’s Technical Guide, Schomäcker et al. [28] determined that a face velocity of 80–120 cfm^−1^ for chemical fume hoods is important for a good working environment. However, in this study, the recommendation was far from what was obtained at the time of sampling. The present study reported 20 cfm^−1^, while the maximum was 41.5 cfm^−1^. It was also documented that an efficient LVS reduces the gaseous and irritant density in a medium environment, especially in a closed system of an environment. Hence, this lessens the harmfulness of chemicals being absorbed under the skin [29,30]. The high airflow in a susceptive environment lessens the odds of erythema among the workers. General ventilation is important in maintaining employee comfort and health in the laboratory and for removing contaminants that would be difficult to contain within a local exhaust hood [5].

### Strengths and Limitations of the Study

The present study has been met with quite a number of limitations due to circumstances beyond our control. One limitation is with respect to the difference in the meteorological conditions in the participants’ study center in the south and north of Nigeria; misclassification was thus observed as recall bias. Incorrectly classifying participants as having erythema using our measurement tool, therefore, could be considered as a limitation. In addition, there was no provision for follow-up of the participants who showed a positive reaction with erythema inducement, and thus the generalizability of the data to a more diverse sample may not be strong. The present study is a cross-sectional study, and thus it is objectively known that the study can only provide a snapshot of the frequency of illness or supplementary physical condition related to other variables with characteristics of interest in a population at a given point in time. This could be considered as a bias, which appears to challenge the positive credential of the documented data.

In spite of the challenges and limitations highlighted, the study has several strengths to be acknowledged. One strength was the high participation rate. The response rate is altogether high when compared to other studies with respect to occupational skin conditions [28].

## 6. Conclusions

The results of the study showed that erythema inducement was significantly associated with and influenced by occupational and environmental factors. Providing better educational knowledge and improving the attitudes towards hazards and safety in a laboratory would lead to a reduced rate of new cases. Moreover, to reduce the dermal absorption and direct contact of these chemical factors, we recommend the proper use of the polymer of ethylene vinyl alcohol co-polymerizate with polyethylene (gloves) as an effective shield against the dermal absorption of these chemical haptens.

## Figures and Tables

**Figure 1 ijerph-16-01334-f001:**
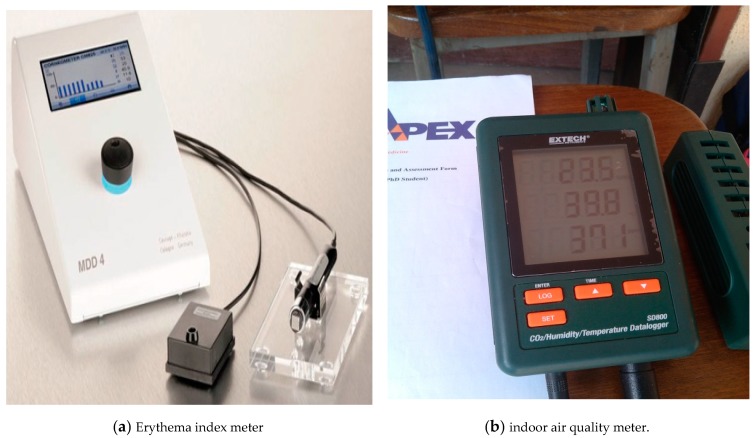
Erythema index meter and indoor air quality meter.

**Figure 2 ijerph-16-01334-f002:**
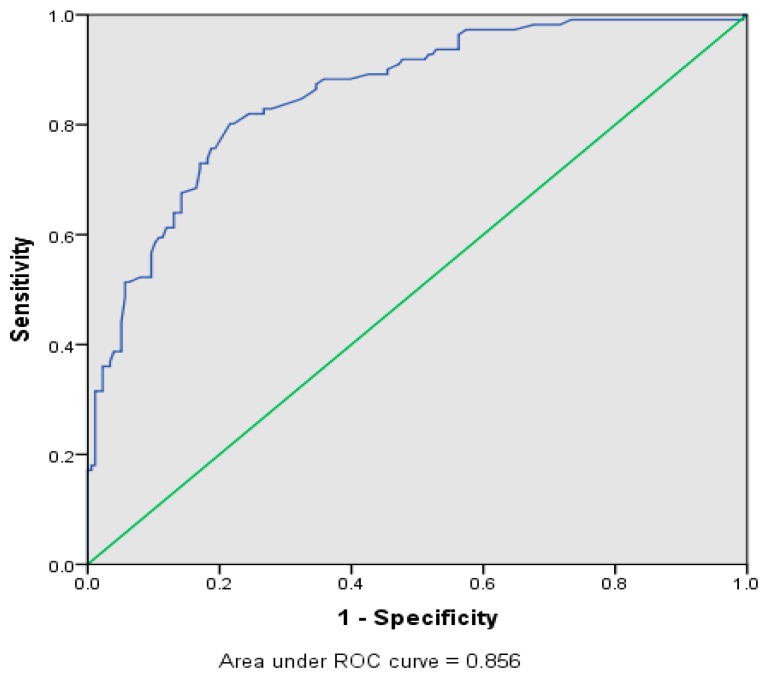
Operating characteristics curve for the fittingof the model(ROC).

**Table 1 ijerph-16-01334-t001:** Characteristics of respondents as well as environmental and chemical parameters concerning the investigated participants (*n* = 287).

**(a) Socio-demographic data**							
**Personal parameters (*n* = 286)**	**Sex**		**Age (Years)**		**Monthly Income (US$)**	**Working Experience (Years)**	
	**Male**	**Female**	**Male**	**Female**			
***n* (%)**	165 (57.7)	122 (42.7)					
**Mean (SD)**			43.1 (8.1)	38.4 (5.1)	246 (120.8)	13.6(6.2)	
**(b)**							
**Chemical parameters (*n* = 30)**	**CO (ppm)**		**CO_2_ (ppm)**		**NO_2_ (ppm)**	**H_2_S (ppm)**	**SO_2_ (ppm)**
**Mean (SD)**	17.9 (2.3)		473.0(52.4)		5.5(0.5)	5.7(0.5)	6.3(0.7)
**(c)**							
**Environmental parameters (*n* = 30)**	**Temperature °C**		**LVS (cfm^−1^)**		**RH (%)**	**LD (m^2^)**	
**Mean (SD)**	28.8(1.5)		25.8(12.6)		24.4 (1.9)	44.7(21.3)	

LVS: laboratory ventilation system; RH: Relative humidity; LD: Laboratory dimension.

**Table 2 ijerph-16-01334-t002:** Occupational factors of skin allergies concerning investigated participants (*n* = 287).

Variables	Skin Allergies–Erythema Inducement		MeanDifference(95% CI)	*t* (df)	χ^2^ (df)	*p*-Value
	Positive Skin Allergy (*n* = 176) *n* (%), Mean (SD)	Negative Skin Allergy (*n* = 111) *n* (%), Mean (SD)				
**PPE**						
Not used	84 (44.0)	107 (56.0)			72.43(1)	<0.001 ^a^
Used	92 (95.8)	4 (4.2)				
**PEL (ppm)**						
Not exceeded	169 (60.4)	111 (39.6)			4.33(1)	0.031 *^,b^
Exceeded	7 (100)	0 (0.0)				
**Con. (mol.dm^3^)**						
Not exceeded	21 (91.3)	2 (8.7)			9.62(1)	0.001 *^,b^
Exceeded	153 (58.4)	109 (41.6)				
Exposed population	528.9 (144.0)	232.3 (106.1)	296.6 (327.8, −265.4)	0.71 (285)		0.001 *^,c^
**Type of chemicals**						
IC and CC > 50%	94 (100)	0 (0.0)			14.1 (2)	0.231 ^a^
IC and CC < 50%	50 (98.0)	1 (2.0)				
HRC and UC > 50%	19 (28.4)	48 (71.6)				
HRC and UC < 50%	13 (17.6)	61 (82.4)				
**Time of exposure(h)**	4.36 (0.70)	3.41 (0.62)	0.953 (1.113, 0.792)	1.14 (285)		<0.001 ^c^

^a^ Pearson’s chi-squared test; ^b^ Fisher’s exact test; ^c^ Independent-sample *t*-test; * < 0.05., PPE: personal protective equipment; PEL: permissible exposure limit; HRC: highly reactive chemicals; UC: unstable chemicals; IC: irritant chemicals; CC: corrosive chemicals.

**Table 3 ijerph-16-01334-t003:** Environmental factors of skin allergies (*n* = 287).

Variables	Skin Allergies–Erythema Inducement		Mean Difference (95% CI)	*t* (df)	χ^2^ (df)	*p*-Value
	Positive Skin Allergy (*n* = 176) *n* (%), Mean (SD)	Negative Skin Allergy (*n* = 111) *n* (%), Mean (SD)				
Temperature (°C)	36.23 (2.15)	32 (3.36)	3.88 (4.36, −3.39)	1.23 (285)	86.1 (1)	<0.001 ^c^
**Laboratory temperature level (°C)**						
Moderate	3 (5.4)	53 (94.6)				<0.001^a^
Poor	173 (74.9)	58 (25.1)				
Relative humidity (%)	34.60 (7.13)	20.32 (7.03)	14.27 (15.9, −12.5)	1.14 (285)	29.3 (1)	<0.001 ^c^
**Laboratory RH level (%)**						
Poor	31 (27.4)	82 (72.6)			90.2 (1)	<0.001 ^a^
Moderate	145 (83.3)	29 (16.7)				
**Indoor air quality (ppm)**						
Good	1 (11.1)	8 (88.9)			24.1 (1)	0.001 *^,b^
Poor	58 (71.6)	23 (28.4)				
Laboratory dimensions(m^2^)	32.02 (11.07)	58.58 (20.20)	26.53 (22.91, 30.19)	0.81 (285)	13.3 (1)	<0.001 ^c^
**Laboratory dimensions (m^2^)**						
Poor	172 (82.3)	37 (17.7)			98.6 (2)	<0.001^a^
Moderate	2 (4.0)	48 (96.0)				
Good	2 (7.1)	26 (92.9)				
**Fume cupboard system**						
Maximum	2 (4.0)	48 (96.0)			96.3 (2)	0.001*
Moderate	12 (16.4)	61 (83.6)				
Minimum	162 (98.8)	2 (1.2)				

^a^ Pearson’s chi-squared test; ^b^ Fisher’s exact test; ^c^ Independent-sample *t*-test; * <0.05.

**Table 4 ijerph-16-01334-t004:** Final model summary and associated factors for erythema inducement (*n* = 287).

Variables		Simple Logistic Regression				Multiple Logistic Regression		
	B	LR/Wald	COR (95%Cl)	*p*-Value	B	LR/Wald	AOR (95%CI)	*p*-Value
**PPE**								
Not used	0		1		0		1	
Used	−1.23	25.34	0.29 (0.12, 0.97)	<0.001	−0.91	18.24	0.40 (0.22, 0.77)	0.001
**PEL (ppm)**								
Not exceeded	0		1		0		1	
Exceeded	1.16	9.11	3.29 (1.02, 9.22)	0.003 *	3.19	4.11	4.22 (2.88, 12.11)	0.004
**TOE (hours)**								
2–3	0		1		0		1	
4–5	1.05	6.32	2.88 (1.00, 7.11)	0.001	2.01	3.55	3.11 (1.77, 9.23)	0.001
**Air laboratory temp (°C)**								
26.6–31.9	0		1		0		1	
≥32	1.95	11.24	7.06 (3.53, 14.05)	0.002 *	2.10	4.82	8.21 (4.03, 15.01)	0.001
**Ventilation (cfm^−1^)**								
≤20	0		1		0		1	
21.5–40.5	−4.38	2.33	0.01 (0.001, 0.05)	<0.001	−3.07	0.22	0.05 (0.004, 0.05)	0.111
≥41.5	−2.88	3.87	0.06 (0.01, 0.24)	<0.001	−1.68	2.53	0.18 (0.02, 0.48)	0.002

* <0.05.Cut-off points, GEV(ACH) = 320 cfm^−1^; Cut-off points, RH = 35–50%, Criteria for FCS = 3 per 90 m^2^; Cut-off points, air laboratory temperature(ALT) = 26.5 °C. TOE: time of exposure; PEL: permissible exposure limit; PPE: personal protective equipment; AOR: adjusted odds ratio; COR: crude odds ratio’ GEV: general exhaust ventilation; ACH: air change per hour; FCS: fume cupboard system.

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
