# Peer review of "Occupational and Environmental Risk Factors Influencing the Inducement of Erythema among Nigerian Laboratory University Workers with Multiple Chemical Exposures"

_ijerph, 2019, doi:10.3390/ijerph16081334_

Round 1
Reviewer 1 Report
Article: ijerph-458564-3
Occupational and environmental risk factors influencing the inducement of erythema among Nigerian laboratory university workers in multiple chemicals exposures
Dear Editor,
Thank you for sending me this article to re-review. Unfortunately, this paper still has issues around the writing. I do not have time to point them all out, and as the authors have assured me twice that they are getting advice on writing and proof reading I am finding the remaining errors frustrating.
Abstract
1. The authors need to rephrase/correct/check the sentence that contains the phrase “Significant erythema”; and “However, a higher level of 37 ventilation reduced the likelihood of erythema in a laboratory by 82% (0.18).”
2. In the paragraph from line 101, technically the Independent t-test should be the Independent samples t-test; the chi-square test is the chi-square test of independence.
3. Your proof reader should have picked up (for example): the tense is wrong for an academic manuscript. The discussion should be past tense. The aim of the study was to…
There are still some very clumsy sentences that should be rephrased.
Throughout the manuscript the space between the full stop and the beginning of the next sentence seems to be missing (eg…. place[1].A…)
line 191 and 192 still have small p’s
Author Response
Dear Reviewer!
We would like to take this opportunity to thank you for the effort and expertise, detailed comments and suggestions for the manuscript, without which it would be impossible to maintain the high standards of the peer-reviewed journal. We believe that the comments have identified important areas which required improvement. After completion of the suggested edits, the revised manuscript has benefited from an improvement in the overall presentation and clarity. Below, you will find a point by point description of how each comment was re-addressed in the manuscript. The original reviewer comments in regular typeface, responses in red ink.
Response to Reviewer 1 Comments (Latest)
Point 1: The authors need to rephrase/correct/check the sentence that contains the phrase “Significant erythema”; and “However, a higher level of 37 ventilation reduced the likelihood of erythema in a laboratory by 82% (0.18).”
Response 1: The authors have revised, rephrased, corrected and clarified accordingly, and believe that it should be much more understandable now.
Point 2: In the paragraph from line 101, technically the Independent t-test should be the Independent samples t-test; the chi-square test is the chi-square test of independence.
Response 2: Corrected accordingly and we believe that it should be much more comprehensible now.
Point 3: Your proofreader should have picked up (for example): the tense is wrong for an academic manuscript. The discussion should be past tense. The aim of the study was to…
There are still some very clumsy sentences that should be rephrased.
Throughout the manuscript, the space between the full stop and the beginning of the next sentence seems to be missing (eg…. place[1].A…)
line 191 and 192 still have small p’s
Response 3: We have revised and corrected all accordingly and believe that it should be much more understandable now.
Thank you for your consideration.
All the above changes have been reflected in the manuscript in yellow text and the latest in pale green text
We hope you find the revised manuscript acceptable for publication. Thank you once again for your consideration.
Yours Sincerely,
Assoc Prof Dr Aziah Daud (MMC No: 31908)
Head of Department
Department of Community Medicine
School of Medical Sciences
Universiti Sains Malaysia Health Campus
16150 Kubang Kerian
Kelantan
Malaysia
Phone : +609 767 6633 / 6622
Fax : +609 767 6654
Email : [email protected]

Reviewer 2 Report
The authors appear to have addressed the reviewers comments satisfactorily.
Author Response
Dear Reviewer!
We would like to take this opportunity to thank you for the effort and expertise, detailed comments and suggestions for the manuscript, without which it would be impossible to maintain the high standards of the peer-reviewed journal. We believe that the comments have identified important areas which required improvement.
Finally, we thank you for being satisfied with the corrections made
Thank you for your consideration.
We hope you find the revised manuscript acceptable for publication. Thank you once again for your consideration.
Yours Sincerely,
Assoc Prof Dr. Aziah Daud (MMC No: 31908)
Head of Department
Department of Community Medicine
School of Medical Sciences
Universiti Sains Malaysia Health Campus
16150 Kubang Kerian
Kelantan
Malaysia
Phone : +609 767 6633 / 6622
Fax : +609 767 6654
Email: [email protected]
This manuscript is a resubmission of an earlier submission. The following is a list of the peer review reports and author responses from that submission.
Round 1
Reviewer 1 Report
Review of
Occupational and environmental risk factors influencing the inducement of erythema among Nigerian laboratory university workers in multiple chemical exposures
Dear Editor,
Thank you for sending me this manuscript to review. This study examines factors associated with sensitivity to chemical agents in laboratories. While this study will most likely be of interest to readers who work in laboratories, the manuscript needs a major revision before it would be suitable for publication.
This manuscript was frustrating to read because the authors used many acronyms without defining them first. For example PEL on line 32, page 1. This issue needs to be addressed throughout the whole manuscript.
Page 2, line 54 needs a reference for the number of injuries.
Page 3, line 99. Were only associations between occupational and environmental characteristics analysed? What about gender etc? Also, nowadays gender should be changed to sex.
What does the author mean by a susceptive environment? Perhaps this sentence could be reworded.
Are there any sensitivity or specificity statistics available for the measurement tool? Had the authors considered misclassification bias? This issue should probably be addressed in the Discussion.
Table 1 and Table 2 are placed in the manuscript, with no accompanying text. These tables need to be introduced and described.
Table 3 fume cupboard system listing needs to be checked that the lables are correct: Not efficient, not so efficient, not at all efficient do not seem correct.
In Table 3 is the order of the variables in lowest to highest or something else meaningful? Currently the order seems a bit haphazard.
The Discussion has no section for strengths or weaknesses of the study.
The Discussion is overly long and just seems like a list of other studies, but I cannot see why the are relevant.
The English needs some editing/proofreading.
Author Response
Response to Reviewer 1 Comments
Point 1: Usage of acronyms in the manuscript by the author without defining them first. For example PEL on line 32, page 1. This issue needs to be addressed throughout the whole manuscript.
Response 1: All acronyms used in the manuscript were now defined first. PEL (personal protective equipment) on line 32, page 1 and other related issues have been addressed accordingly throughout the whole manuscript.
Point 2: Reference is required for the number of injuries from chemical exposure in the laboratory on page 2, line 54.
Response 2: Reference is provided as …..…Ekwempu., et al, 2018. for the number of injuries from chemical exposure in the laboratory on page 2, line 54 and all other related issues have been corrected accordingly throughout the whole manuscript.
Point 3a: A question asked whether only associations between occupational and environmental characteristics were analyzed on page 3, line 99 in the study and not captured gender.
Response 3a: Thank you, Sir, for this very important issue rose. However, from the conceptualization and the design of the study, it looks like the three variables; occupational environmental, and socio-demography i.e. sex may not be compatible in one model. During the preliminary analyses, socio-demography such as sex was statistically insignificant. Also, in the final model, the Receiver Operating Characteristics (ROC) was very low, and large P-value, as well as a wide confidence interval, were observed which made us to report only two variables as capture in the tile wihle socio-demography i.e. sex was considered for only descriptive purpose
Point 3b: Suggestion to change gender to sex in Table 1 on page 3
Response 3b: Gender as socio-demography factor has been replaced as sex in Table 1 on page 3 as suggested and corrected throughout in the whole study.
Point 4: A question asked, what it meant by a susceptive environment described by the author in the text, perhaps the sentence could be reworded as highlighted by the reviewer.
Response 4: Susceptive environment in the study is meant, as an environment that is likely to be influenced or induced erythema among the workers or easily influenced by poor educational knowledge and attitude of hazards and safety in a laboratory.
Point 5a: A question asked whether sensitivity or specificity statistics are available for the measurement tool in the study.
Response 5a: The statistical measurement of binary classification test in SPSS was not performed directly, however, it was only summarized in the model as model stability and the diagnostic ability of the binary classifier which is now shown in Figure 2 and were found to be 85.6% thus, showing the model to be fit as a classification function often known as (true positive rate) or (true negative rate).
Point 5b A question asked if authors had considered misclassification bias. That the issue should probably be addressed in the Discussion.
Response 5b: Misclassification bias has now been considered as the limitation of the study and has been discussed under the strengths and limitation section in the manuscript.
Point 6: Table 1 and Table 2 are placed in the manuscript, with no accompanying text. It required that these tables need to be introduced and described.
Response 6: Table 1 and Table 2 have now been introduced and properly describe accompanying with text in the manuscript.
Point 7: Required that the fume cupboard system listing in Table 3 needs to be checked that most likely labels do not seem correct i.e. not efficient, not so efficient, not at all efficient
Response 7: Thank you, Sir, for this observation. Nevertheless, we have discussed this observation as a team and seem like the classification and the labeling is correct. ASHRAE and CLEAPSS recommendation standard that a laboratory fume cupboard system per standard chemical laboratory (ACH-cfm-1 >90m2) during a working session (LFS) <1(PSL, 90m2); LFS Not at all efficient (NAAE) =0, 1(PSL, 90m2); LFS not so efficient (NSE) =1,>1(PSL, 90m2); LFS very efficient (VE) =2. PSL. However, we would appreciate if you can assist in improving and strengthening the classification and the labelling. (Note; PLS= per standard laboratory).
Point 8: In Table 3, it is required to know if the order of the variables in lowest to highest or something else meaningful? That the current order seems a bit haphazard.
Response 8: The variables in Table 3 were ordered preferential to the best selection procedure and the preliminary main effect model processed using the enter method. The model accounted for the matching by factors best on the statistically significant variables
Point 9: The Discussion has no section for the strengths or weaknesses of the study.
Response 9: Strengths and limitation of the study have been discussed under the strengths and limitation section in the manuscript.
Point 10: The Discussion is overly long and just seems like a list of other studies, but I cannot see why are relevant.
Response 10: Thank you, Sir, for this observation, though, we thought we could summarize the discussion to look better, relevant and captivating than the current one
Point 11: It requires that English needs some editing/proofreading
Response 11: The whole manuscript now has been edited and proofread by English-language editing service, to be precise by MDPI Author Services

Reviewer 2 Report
The paper deals with an interesting and useful topic for readers. However, there are major changes required before the paper would be ready to publish including:
The methods were not adequately explained in particular how the gas readings were carried out (specific details of equipment used, if equipment was within calibration, etc.) and more details around the statistical analysis is required.
The discussion section was weak and needs more work.
The abstract was poor and needs work.
The conclusion should include some detailed recommendations for practice based on the findings of the study.
The written English is poor and it was difficult to read the paper.
Author Response
Response to Reviewer 2 Comments
Point 1a: The methods were not adequately explained in particular how the gas readings were carried out (specific details of equipment used
Response 1a: The method on how gas readings were carried out has now been adequately explained (see page 3, line 120-131 under measurement tool). For instance; Dosimeter tube gases of interest also known as NEXTTEQ 7446-09-5 [12] and IAQCM was calibrated, configured, and equipped with a sensor to measure the different level of carbon dioxide (CO2), carbon monoxide (CO), temperature (oc), relative humidity (%). However, sulphur dioxide (SO2), hydrogen sulphide (H2S) and nitrogen dioxide (NO2) were measured using dosimeter tubes gases of interest at Time Weighted Average (TWA). These tubes are equipped with a length-of-stain indication proportional to the amount of gas contaminant of interest present in the laboratory, ending with a discrete line of differentiation. The value on the scale that corresponds to the end of the stain length was the concentration of the target gas. The average concentrations of the gases were obtained by dividing the reading by the total length of time that the tube was exposed (expressed in hours) in the laboratory as seen in the expression below;
Point 1b: Required a specific details of equipment used.
Response 1b: The equipment used in the study include; erythema index meter (EIM) which also known as MX18, dosimeter tube gases of interest also known as NEXTTEQ 7446-09-5 and indoor air quality control meter (IAQCM) which is also known as EXTECH MODEL SD800
Point 1c: Required to know if equipment was within calibration,
Response 1c: Yes, Sir, the equipment were within calibration (calibrated by team of expert in the field)
Point 1d: more details around the statistical analysis are required.
Response 1: additional statistical analyses have been provided (see page 5, line 160-172). For instance; the x2test revealed personal protective equipment (PPE), permissible exposure limit (PEL), air laboratory temperature (ALT), indoor air quality IAQ and laboratory ventilation system (LVS) was significantly associated with erythema inducement with p<0.001, P=0.031, p<0.001, P=0.001, and P=0.002 respectively (Table 2). Also, independent sample t-test revealed that exposed population and time of exposure (TOE) have significant difference between these two groups with mean difference (95% CI)), p-value = 0.001 and p<0.001 respectively (Table 3). In addition, the Receiver Operating Characteristics (ROC) described the model stability and the diagnostic ability of the binary classifier showed in Figure 2 were found to be 85.6% thus, showing the model to be fit
Point 2: The discussion section was weak and needs more work.
Response 1: Few data have been added in the manuscript, as well as strengths and limitation of the study. For instance the Receiver Operating Characteristics (ROC) described the model stability and the diagnostic ability of the binary classifier showed in Figure 2 was found to be 85.6% thus, showing the model to be fit. The discussion has now been strengthened and seems captivating.
Point 3: The abstract was poor and needs work
Response 3: Few data have been added in the abstract and was re-introduced systematically and now seems better and organised.
Point 4: The conclusion should include some detailed recommendations for practice based on the findings of the study.
Response 4: The conclusion has been improved including details recommendations for practice based on the findings of the study (see the conclusion section). For instance “to reduce the dermal absorption and direct contact of these chemical factors, we recommend the proper use of the polymer of the ethylene vinyl alcohol co-polymerizate with polyethylene (gloves) for an effective shield against dermal absorption of these chemical haptens”
Point 5: The written English is poor and it was difficult to read the paper
Response 5: The whole manuscript now has been edited and proofread by English-language editing service, to be precise by MDPI Author Services.

Round 2
Reviewer 1 Report
Review for: Manuscript ID ijerph-458564
Occupational and environmental risk factors influencing the inducement of erythema among Nigerian laboratory university workers in multiple chemicals exposures
Dear Editor,
Thank you for sending me this article to re-review. Unfortunately, there are still a number of issues that need addressing.
1. As your reviewer is a female, I don't appreciate being addressed as "Sir". Further paraphrasing my review comments, makes it very unclear whether you have actually done what I have suggested.
2. The English editing needs to be rechecked. For example, Page 3 , line 95 reads "2years" not 2 years. There is still a lot of wavering between the acronym and the fully written term. P-values are reported inconsistently as P and p. Sentences should not be started with an acronym (eg PEL...).
3. Please not the term for statistical non-significance is not "insignificance"
4. The Results report the use of a Chi-square test (and a t-test), however, there is no mention of these tests in the statistical methods. The Chi-square test should be properly reported in the Results section...Chi-square test of Independence (or whatever it was, not just the sumbol).
5. Regarding Figure 2, I think that you mistakenly think I was asking for the sensitivity and specificity of the statistical models? What I wanted to know was how accurate (sensitive and specific) the machine was that you were using to classify erythema (see Measurement tools section).
6. Regarding Point 7.
"Point 7: Required that the fume cupboard system listing in Table 3 needs to be checked that most likely labels do not seem correct i.e. not efficient, not so efficient, not at all efficient.
Response 7: Thank you, Sir, for this observation. Nevertheless, we have discussed this observation as a team and seem like the classification and the labelling is correct. ASHRAE and CLEAPSS recommendation standard that a laboratory fume cupboard system per standard chemical laboratory (ACH-cfm-1 >90m2) during a working session (LFS) <1(PSL, 90m2); LFS Not at all efficient (NAAE) =0, 1(PSL, 90m2); LFS not so efficient (NSE) =1,>1(PSL, 90m2); LFS very efficient (VE) =2. PSL. However, we would appreciate if you can assist in improving and strengthening the classification and the labelling. (Note; PLS= per standard laboratory). "
I am confused, the classification you list in your reply is different to the classification in your Table.
7. Regarding Point 8.
“Point 8: In Table 3, it is required to know if the order of the variables in lowest to highest or something else meaningful? That the current order seems a bit haphazard.
Response 8: The variables in Table 3 were ordered preferential to the best selection procedure and the preliminary main effect model processed using the enter method. The model accounted for the matching by factors best on the statistically significant variables: …”
Perhaps this point could be made somewhere in the manuscript?
8. The limitations section is very weak, given that I think you have misinterpreted my point about the sensitivity and specificity of your measurement tool to classify erythema. The limitation is that you may have incorrectly classified people as having erythema (using your measurement tool).
9. The Discussion still needs work. The Discussion needs to restate the purpose, and findings of the study; then the authors need to match the study findings with other studies which do (or do not) support the current study findings. This Discussion still reads like a list of other studies and it is very difficult to see the point in reading what is written.
Reviewer 2 Report
Thank you for adding details regarding your sampling methods and equipment for hazardous chemicals in the laboratory air. It is much clearer now what was done. However, you need to make a minor correction. In the paper you state:
"However, sulfur dioxide 126 (SO2), hydrogen sulfide (H2S) and nitrogen dioxide (NO2) were measured as Time Weighted Averages 127 (TWA) using dosimeter tubes containing the gases of interest.."
The dosimeter tubes do not contain "the gases of interest". They contain substances that have a reaction with the gas of interest, producing a colour change.
This information needs to be corrected.